# Lung microRNAs Expression in Lung Cancer and COPD: A Preliminary Study

**DOI:** 10.3390/biomedicines11030736

**Published:** 2023-02-28

**Authors:** Davida Mirra, Renata Esposito, Giuseppe Spaziano, Chiara La Torre, Cristina Vocca, Martina Tallarico, Erika Cione, Luca Gallelli, Bruno D’Agostino

**Affiliations:** 1Department of Environmental Biological and Pharmaceutical Sciences and Technologies, University of Campania “Luigi Vanvitelli”, 81100 Caserta, Italy; 2Department of Pharmacy, Health and Nutritional Sciences-Department of Excellence 2018–2022, University of Calabria, 87036 Rende, Italy; 3Clinical Pharmacology and Pharmacovigilance Unit, Department of Health Sciences, Mater Domini Hospital, University of “Magna Graecia”, 88100 Catanzaro, Italy

**Keywords:** microRNAs, lung tissue, COPD, lung cancer

## Abstract

Non-small cell lung cancer (NSCLC) is one of the deadliest diseases worldwide and represents an impending burden on the healthcare system. Despite increasing attention, the mechanisms underlying tumorigenesis in cancer-related diseases such as COPD remain unclear, making novel biomarkers necessary to improve lung cancer early diagnosis. MicroRNAs (miRNAs) are short non-coding RNA that interfere with several pathways and can act as oncogenes or tumor suppressors. This study aimed to compare miRNA lung expression between subjects with NSCLC and COPD and healthy controls to obtain the miRNA expression profile by analyzing shared pathways. Lung specimens were collected from a prospective cohort of 21 sex-matched subjects to determine the tissue miRNA expression of hsa-miR-34a-5p, 33a-5p, 149-3p, 197-3p, 199-5p, and 320a-3p by RT-PCR. In addition, an in silico prediction of miRNA target genes linked to cancer was performed. We found a specific trend for has-miR-149-3p, 197-3p, and 34a-5p in NSCLC, suggesting their possible role as an index of the tumor microenvironment. Moreover, we identified novel miRNA targets, such as the Cyclin-Dependent Kinase (CDK) family, linked to carcinogenesis by in silico analysis. In conclusion. this study identified lung miRNA signatures related to the tumorigenic microenvironment, suggesting their possible role in improving the evaluation of lung cancer onset.

## 1. Introduction

Non-small cell lung cancer (NSCLC) is a deadly disease and is often not diagnosed until an advanced stage, leading to poor overall survival and representing an impending burden on healthcare systems worldwide [1]. It consists of the uncontrolled growth of abnormal cells in one or both lungs, interfering with normal cell functions and the lungs’ capacity to provide oxygen to the body [2,3,4]. Among the numerous cancer-related diseases [5,6,7,8,9,10], the most common is chronic obstructive pulmonary disease (COPD), a progressive inflammatory disease characterized by long-term respiratory symptoms and incompletely reversible airflow limitation [11,12,13]. NSCLC and COPD share several features, such as underlying common inflammatory processes and environmental exposure [14,15,16]. Indeed, for a long time, the high rate of NSCLC in patients with COPD was thought to reflect smoke exposure, the most common etiological risk factor [17,18,19]. However, only some smokers develop either disease (or both), highlighting the role of genetic susceptibility to disease [20,21]. Therefore, the link between these two diseases has prompted in-depth epidemiological studies to address this issue [22,23,24]. Despite this increased attention, the mechanistic nature remains unclear, making novel biomarkers necessary to improve NSCLC early diagnosis. In previous studies, we have reported that microRNAs (miRNAs) are correlated with several diseases [25,26,27,28]. MicroRNAs (miRNAs) are a family of short non-coding RNA (approximately 18–22 nucleotides) that act as post-transcriptional regulators through silencing of their downstream target mRNA [29,30]. Emerging evidence suggests an association of miRNAs with the augmentation and amelioration of NSCLC by interfering with several regulatory pathways and acting as oncogenes or tumor suppressors, as demonstrated by remarkable changes in gene expression and protein synthesis [31,32]. Indeed, several miRNAs were found to be involved in the regulation of proliferative signaling, evasion of growth suppressors, activation of invasion and metastasis, enablement of replicative immortality, angiogenesis, and resistance to programmed cell death [33]. In the present study, we focused on miRNAs that have been shown to influence cancer hallmarks, such as cell proliferation and immunosuppression [34,35,36,37,38]. Notably, has-miR-34a-5p is a tumor suppressor that targets the epidermal growth factor receptor (EGFR) and is often lost or reduced in NSCLC [39]. Hsa-miR-33a-5p, which directly targets myeloid cell nuclear differentiation antigen (MNDA), a member of the hematopoietic protein family, was found to be upregulated in lung adenocarcinoma and was associated with poorer overall survival [40]. Similarly, has-miR-197-3p was reported to be overexpressed in NSCLC tissues and cells compared with normal tissues and cells [41]. However, even if several miRNAs are crucially involved in NSCLC, a specific miRNA signature is still needed because miRNA profiles could be common in COPD and cancer. Indeed, several miRNAs targeting candidate mRNA families, such as cytokines, antioxidant enzymes, proteinases, and anti-proteinases, have been implicated in both diseases’ pathogenesis and may be of prognostic value, giving a proportion of the risk [42]. Therefore, miRNA assessment and determination of their positions in the regulatory pathways are of great interest for improving the diagnosis and evaluation of susceptibility to NSCLC in patients with COPD. Therefore, we analyzed an miRNA panel in subjects with NSCLC and COPD and healthy individuals to obtain common miRNA signatures between COPD and NSCLC, improving the diagnosis and evaluation of susceptibility to NSCLC in patients with COPD.

## 2. Materials and Methods

### 2.1. Study Population

We performed an observational clinical study on 21 age- and sex-matched subjects aged 18 years or older who were referred to the “Mater Domini” Hospital in Catanzaro, Italy, with a suspected diagnosis of pulmonary neoplasia and were undergoing routine bronchoscopy. This study was part of a clinical trial recorded at clinicaltrials.gov (NCT04654104) and was approved by the local ethics committee “Calabria Centro”. This study was conducted in compliance with the Institutional Review Board/Human Subjects Research Committee, the Declaration of Helsinki, and the Guidelines for Good Clinical Practice criteria. Before the beginning of the study, all the enrolled patients or legal guardians signed the informed consent. Patients’ demographics and clinical and social histories were obtained at enrollment. All subjects underwent spirometry according to international guidelines [43]. The exclusion criteria were active pulmonary infections; autoimmune diseases; extrapulmonary neoplasia; and other airflow obstructions, such as asthma or bronchiectasis, and those who did not sign the informed consent form at the time of enrollment were also excluded. The enrolled subjects were then classified into three groups according to their clinical and pathological (bronchoscopy-guided) diagnoses: (1) a healthy control group (“HC”; n = 6); (2) a non-small cell lung cancer group (“NSCLC”; n = 9); and (3) subjects with COPD (“COPD,” n = 6).

### 2.2. Data Collection and Clinical Biochemistry Assays

The clinical materials accessible for the study were archival formalin-fixed and paraffin-embedded (FFPE) lung specimens. FFPE tissues arrived sectioned at a 20 μm thickness. For the extraction of microRNAs (total RNA) from FFPE tissue blocks, we adopted the best method available, the miRNeasy FFPE kit [44]. Briefly, each block was placed in 320 µL of deparaffination solution, incubated for 3 min at 56 °C in a thermomixer (Eppendorf, Hamburg, Germany), cooled at room temperature, and placed in 240 µL of lysis buffer with 10 µL of proteinase K. Digestion was started for 15 min at 56 °C, followed by reverse formalin crosslinking and denaturing the enzyme, and the temperature was increased to 85 °C for a further 15 min. Genomic DNA was then rapidly removed in an optimized DNase step with 25 µL of DNase in Booster Buffer at room temperature for 15 min, and after several washes with RBC buffer and RPE buffer, the concentrated RNA was purified using RNeasy MinElute spin columns. RNA was eluted in a volume of 15 µL. The qubit RNA Integrity and Quality (IQ) assay (catalog number Q33222) was performed to check RNA degradation using a Qubit 4 fluorometer (serial number 2322618032114).

### 2.3. Real-Time PCR (RT-PCR)

The expression miRNA levels of hsa-miR-34a-5p, 33a-5p, 149-3p, 197-3p, 199-5p, and 320a-3p was performed with a method termed looped primer RT-PCR, following Thermo Fisher Scientific (Waltham, MA, USA) [45].

The sequences used were:hsa-miR-34a-5p: GGCCAGCUGUGAGUGUUUCUUUGGCAGUGUCUUAGCUGGUUGUUGUGAGCAAUAGUAAGGAAGCAAUCAGCAAGUAUACUGCCCUAGAAGUGCUGCACGUUGUGGGGCCC (Catalog number 478048_mir);hsa-miR-33a-5p: CUGUGGUGCAUUGUAGUUGCAUUGCAUGUUCUGGUGGUACCCAUGCAAUGUUUCCACAGUGCAUCACAG (Catalog number 478347_mir);hsa-miR-149-3p: GCCGGCGCCCGAGCUCUGGCUCCGUGUCUUCACUCCCGUGCUUGUCCGAGGAGGGAGGGAGGGACGGGGGCUGUGCUGGGGCAGCUGGA (Catalog number 478720_mir);hsa-miR-197-3p: GGCUGUGCCGGGUAGAGAGGGCAGUGGGAGGUAAGAGCUCUUCACCCUUCACCACCUUCUCCACCCAGCAUGGCC (Catalog number 477959_mir);hsa-miR-199a-5p: GCCAACCCAGUGUUCAGACUACCUGUUCAGGAGGCUCUCAAUGUGUACAGUAGUCUGCACAUUGGUUAGGC (Catalog number 478231_mir);hsa-miR-320a-3p: CUCCCCUCCGCCUUCUCUUCCCGGUUCUUCCCGGAGUCGGGAAAAGCUGGGUUGAGAGGGCGAAAAAGGAUG (Catalog number 478594_mir);Additionally, U6 snRNA: GTGCTCGCTTCGGCAGCACATATACTAAAATTGGAACGATACAGAGAAGAT CATGGCCCCTGCGCAAGGATGACACGCAAATTCGTGAAGCGTTCCATATTTT (Catalog number 001973) was selected as housekeeping miRNA.

Briefly, 5–10 ng of total RNA was subjected to reverse transcription polymerase chain reaction using the TaqMan MicroRNA Reverse Transcription kit for all the miRNA targets chosen, according to a manufacturer’s protocol. The thermocycling conditions were as follows: 42 °C for 30 min, 85 °C for 5 min 85 °C, and 4 °C for 5 min. RT-PCR was performed using the TaqMan Universal PCR Master Mix Kit (Thermo Fisher Scientific, Waltham, MA, USA) according to the manufacturer’s protocol and QuantumStudio3^TM^ Real-Time PCR Systems [46]. The thermocycling conditions were as follows: 95 °C for 10 min and 40 cycles of 15 s at 95 °C, followed by 1 min at 60 °C. Six biological replicates for the HC group, nine for NSCLC, and six for COPD were analyzed, and all samples were run in triplicate; after the achievement of the RT-PCR, the cycle threshold (Ct) of the reactions was determined. Data from all RT–PCR experiments and miRNA expression was analyzed by normalizing to the endogenous miRNA control applying the comparative 2^^ΔΔCt^ method, where ΔCt = Ct_miRNA_ – Ct _housekeeping miRNA_, whereas the relative quantification of differences in expression was conducted with ΔΔCt = ΔCt_HC_ − ΔCt_NSCLC/COPD_. 

### 2.4. In Silico Prediction of “hsa-miR” Target Genes

In order to identify genes as targets of hsa-miR-34a-5p, 33a-5p, 149-3p, 197-3p, 199a-5p, and 320a-3p linked to NSCLC and/or COPD, we performed an in silico analysis. The in silico identification of the target genes was performed using the miRTargetLink Human (https://ccb-web.cs.uni-saarland.de/mirtargetlink/, accessed on 11 November 2022) and DIANA Tools (http://diana.imis.athena-innovation.gr/DianaTools/index.php, accessed on 18 November 2022) databases. The latter database was used to check which miRNA target genes were already validated experimentally. Possible biochemical pathways involved were checked using the GeneCard database (https://www.genecards.org/, accessed on 30 November 2022), which allowed genomic, proteomic, transcriptomic, genetic, and functional information on all known and predicted human genes to be obtained.

### 2.5. Statistical Analysis

All data are expressed as means ± standard deviation (SD). The one-way ANOVA test followed by Dunn’s multiple comparison test was used to evaluate the differences between the groups. We used nominal (sex, comorbidity, and treatment) and categorical variables (age, weight, and grade of disease) and correlations between clinical characteristics were calculated using one-way ANOVA followed by the Tukey Multiple Comparison Test. GraphPad software (version 8.0) was used for statistical analyses (GraphPad Software, San Diego, CA, USA). Differences were considered statistically significant at *p* < 0.05.

## 3. Results

### 3.1. Patients

Clinical characteristics and pharmacological treatment are reported in Table 1. Among the subjects presenting cancer, only those with NSCLC were enrolled. The most frequent comorbidities were diabetes and cardiovascular diseases (*p* < 0.0001), and the most common medications used were inhaled corticosteroids and bronchodilators (*p* < 0.001 and *p* < 0.0001 respectively).

### 3.2. miRNA Expression in NSCLC

Total RNA was extracted from paraffin-embedded samples with an extraction efficiency between 82 and 97% for all samples analyzed. The hsa-miR-33a-5p, 149-3p, 197-3p, 199a-5p, and 320a-3punderwent significant positive modulation in LC compared with those in HC, with a *p*-value of <0.0001 (Figure 1A–E). By contrast, the RT-PCR analysis indicated that hsa-miR-34a-5p was significantly reduced in NSCLC compared with that in HC, with a *p*-value of <0.0001 (Figure 1F). 

### 3.3. miRNA Expression in COPD

The RT-PCR analysis indicated that hsa-miR-33a-5p, 199a-5p, 320a-3p, and 34a-5p were significantly increased in subjects with COPD compared with those in HCs, with a *p*-value of <0.0001 (Figure 1A,D–F). By contrast, hsa-miR-149-3p and 197-3p were significantly reduced in COPD compared with those in HC, with *p*-values of <0.0001 and <0.001, respectively (Figure 1B,C). 

### 3.4. miRNA Expression in NSCLC vs. COPD

Determination via RT-PCR analysis indicated that hsa-miR-33a-5p, 149-3p, 197-3p, 199a-5p, and 320a-3p underwent significant positive modulation in NSCLC compared with those in COPD, with a *p*-value of <0.0001 (Figure 1A–E). By contrast, the hsa-miR-34a-5pwas significantly reduced in NSCLC compared with that in COPD, with a *p*-value of <0.0001 (Figure 1F).

### 3.5. In Silico Results

Two different databases were used for the in silico analysis. Data were compared with respect to the number of target genes experimentally validated in both databases. The results are reported in Table 2. In DIANA Tools, the numbers of validated target genes were higher with respect to the miR target Link Human; therefore, DIANA Tools was used for bioinformatics analysis. We analyzed all selected miRNA targets, prioritizing those involved in cellular control-related pathways known to be the main triggers in the phenomenon of carcinogenesis. Abbreviations, names of the selected genes, methods, and tissues of validation are reported in Table 3. The hsa-miR-34a-5pwas found to regulate ADAM10, XBP1, CDK6, and E2F3, which are involved in the NOTCH1 pathway, cell migration and invasion, cycle regulation, and the Bcl 2 signaling pathway, respectively, as shown in Table 4. The hsa-miR-33a-5p and 320a-3p were found to regulate ZNF281, which is involved in the NF-kB pathway and the regulation of cell apoptosis and proliferation, while hsa-miR-199a-5p and 197-3p target MARCH8 and MARCH9, respectively, which regulate the PI3K, mTOR, and ICAM-1 signaling pathways. However, other target genes linked to tumor invasion and metastasis were found to be influenced by hsa-miR-199q-5p, 149-3p 320a-3p, and 33a-5p, as shown in Table 4.

## 4. Discussion

To our knowledge, this is the first study to compare tissue miRNA profiles in NSCLC and COPD patients, attempting to identify specific miRNA signatures that could be involved in tumorigenesis and the underlying mechanisms that make COPD subjects more prone to cancer. COPD is a progressive inflammatory disease characterized by immune system dysregulation and airflow limitation that represents a significant risk factor for NSCLC [11,12,13]. Both diseases share several features, and many patients with COPD develop NSCLC every year, making useful biomarkers to predict this risk necessary. For these reasons, many studies have attempted to reveal the common mechanisms between COPD and NSCLC, but they have only partially answered this question [22,23,24]. Due to the harsh nature of fixation and embedding procedures involved in preserving clinical material, the samples are often heavily fragmented and chemically cross-linked by formalin. miRNAs, due to their small size and increased stability, are easier to retrieve and study in these precious tissues and are becoming extremely clinically relevant [47]. Herein, we performed an analysis of an miRNA panel and investigated common pathways in both diseases, identifying similarities and differences in miRNA tissue profiles that may be of prognostic value by giving a proportion of the risk of cancer development in COPD patients. We found common hsa-miRNA-33a-5p, 199a-5p, and 320a-3p trends and an opposite trend for has-miR-149-3p, 197-3p, and 34a-5p. Specifically, the levels of the first three miRNAs were increased in both LC and COPD patients compared with those in healthy controls. Although our experiments showed high expression of hsa-miR-199a-5p, it is generally recognized as a tumor suppressor and is often downregulated in cancer tissues and cells, leading to the upregulation of many target genes that contribute to tumor progression or radiation and chemotherapy resistance [48]. Among them, we found that hsa-miR-199a-5p could target CAPRIN1, an RNA-binding protein generally overexpressed in various cancer types that acts as a modulator of cell proliferation, immune checkpoint proteins, and resistance to radiation and chemotherapy [49]. However, hsa-miR-199a-5p regulatory effects on tumor progression and pulmonary fibrosis are different and cannot be generalized, making its role controversial [50]. Indeed, in silico analysis also revealed that hsa-miR-199a-5p targets tumor suppressor genes such as MARCH8; thus, its increased expression could induce cancer progression [51,52]. In addition, our data in the COPD group suggested that hsa-miR-199a-5p expression could be indicative of a common pathway with NSCLC; this is in accordance with Mizuno et al., who reported that the upregulation of this miRNA was correlated with HIF-1a protein levels in the lungs of COPD patients [53]. Myeloid cell nuclear differentiation antigen (MNDA) is a nuclear protein that regulates innate immunity, apoptosis, and cancer by interacting with many effector proteins [54]. Tang et al. reported that MNDA expression was lower in lung adenocarcinoma, and this response correlated with the upregulation of has-miR-33a-5p [40]. Furthermore, Wang et al. showed that hsa-miR-33a-5p was overexpressed in lung squamous cell carcinoma tissues compared with that in normal lungs, thus playing an essential role in the onset and development of the disease [55]. Consistent with these data, we observed a significant increase in hsa-miR-33a-5p in the LC group and the COPD group with respect to healthy ones, thus providing evidence of cancer-like dysregulation in COPD. Indeed, despite the lack of literature data referring to this miRNA in COPD patients, Lai et al. reported that it enhanced proinflammatory cytokines induction and NF-kB activation after LPS stimulation [56]. In addition, we found an in silico interaction between hsa-miR-33a-5p and genes such as SMAD7 and ZNF281. Considering that SMAD7 acts as a transcriptional regulator and ZNF281 is a tumor-suppressive long non-coding (lnc)RNA, an increase in hsa-miR-33a-5p could lead to the downregulation of these factors and consequently to a loss of tumor suppression capability [57,58,59]. Among cells contributing to pro-tumor microenvironments, macrophages, especially of the M2 phenotype, are key players due to their ability to infiltrate the tumor microenvironment, leading to cancer progression [60]. Fortunato et al. described an overexpression of hsa-miR-320a-3p in heavy-smoker macrophages, which induced the M2 phenotype through the downregulation of STAT4 [61]. In accordance, our data showed evidence that NSCLC was associated with a significant increase in hsa-miR-320a-3p compared with that in healthy subjects. Moreover, by in silico analysis, we found that hsa-miR-320a-3p could reduce tumor suppression capacity by targeting LATS 1 and 2, factors that regulate organ size and cell proliferation [62,63]. Although this miRNA has never been evaluated in COPD, Matamala et al. reported that the upregulation of has-miR-320c, which belongs to the same family, plays a role in alpha 1 antitrypsin regulation during the pulmonary inflammatory process [64]. Interestingly, NSCLC and COPD exhibited opposite trends in hsa-miR-149-3p, 197-3p, and 34a-5p, suggesting that their dysregulation in subjects with COPD toward levels found in cancer could be a specific index of the pro-tumorigenic microenvironment. An imbalance of protease–antiprotease is a key feature in the pathogenesis of emphysema in patients with COPD. Among these factors, Serpin Peptidase Inhibitor, Clade E Member 1 (SERPINE1) is a protease inhibitor that has been shown to be upregulated in the sputum of patients with severe COPD or during hypoxia [65], exposure to LPS, cigarette smoke [66], and oxidative stress [67]. Savarimuthu Francis et al. described higher SERPINE1 expression in severe emphysema that was inversely correlated with hsa-miR-149-3p, suggesting its implication in the pathogenesis of the disease [68]. Shen et al. reported lower hsa-miR-149-3p blood levels in smokers with stable COPD, smokers with acute exacerbation of COPD, and healthy smokers than in healthy non-smokers, highlighting the possible role of cigarette smoke in regulating this miRNA [69]. Consistent with these data, we observed a significant decrease in hsa-miR-149-3p in COPD with respect to both NSCLC patients and healthy ones. This finding was in accordance with that of Tian et al., who found that hsa-miR-149-3p was highly expressed in NSCLC cells and tissues, promoting cancer proliferation and motility [70]. Furthermore, we found an in-silico interaction between hsa-miR-149-3p and HEATR1, a protein mainly involved in the p53 and mTOR signaling pathways, which was found to be upregulated in NSCLC tissue compared with that in healthy adjacent ones [71]. Our observation of increased levels of hsa-miR-149-3p suggests a downregulation of HTEAR1 in our disease model, making further investigation of their role in NSCLC necessary. Similarly, hsa-miR-197-3p showed an opposite trend between NSCLC and COPD subjects, suggesting an increase in cancer and a decrease in COPD. Metastasis-associated lung adenocarcinoma transcript 1 (MALAT1) is usually overexpressed in deteriorated neoplasms, including NSCLC [72]. Yang and colleagues reported that MALAT1 and has-miR-197-3p were both overexpressed in NSCLC tissues and cells when compared with normal tissues, associating them with promoted growth and chemoresistance of cancer [41]. Moreover, consistent with this, we found an in-silico interaction between has-miR-197-3p and the tumor suppressor MARCH 9, suggesting that its upregulation leads to decreased expression of this gene and contributes to enhanced cancer cell survival [73]. The intimal proliferation of dedifferentiated muscle cells is the main contributor to pulmonary vascular remodeling, a common feature of COPD [74]. According to our data, in the pulmonary arteries of COPD patients, vessel wall remodeling has been associated with the downregulation of hsa-miR-197-3p [75]. Finally, the LC group and COPD patients differed in the has-miR-34a-5p trend, showing an increase and decrease in lung expression compared with healthy controls, respectively. hsa-miR-34a-5p is one of the most interesting miRNAs that has been extensively evaluated in NSCLC [76]. It is commonly considered a tumor suppressor, and many of its validated targets have been implicated in the control of cellular proliferation, apoptosis, and immune evasion [77,78]. Gupta et al. reported that has-miR-34a-5p is the main controller of has-miR-16 activity via HDAC1 and/or c-Myc, thus coordinately regulating the cell cycle checkpoint in NSCLC cells [79]. In addition, it has been shown that oxidative stress could selectively elevate has-miR-34a-5p with a concomitant reduction in SIRT1/-6 in bronchial epithelial cells and peripheral lung samples from patients with COPD, in accordance with our findings [80]. Interestingly, we observed an in-silico interaction between hsa-miR-34a-5p and CDK6, a protein kinase belonging to the CDK family. CDKs are involved in cell cycle control, and their dysregulation can lead to uncontrolled cell growth [81]. Specifically, CDK6 regulates the progression of the cell cycle from the G1 to the S phase, and its expression is often increased in a variety of cancers [82]. Since hsa-miR34a-5p is downregulated in LC, this dysregulation may lead to aberrant expression of CDK6 with uncontrolled cell proliferation. Similarly, we found that hsa-miR-34a-5p targets ADAM10, XPB1, and E2F3, all of which contribute to cell migration and invasion [83,84,85,86]. It is important to point out that miRNA expression profiles could be confounded by environmental factors, such as cigarette smoke. Several miRNAs have been associated with smoke exposure, acting as mediators of smoking-induced inflammation and target organ damage [87]. Moreover, smoking status could strictly depend on gender. Indeed, men tend to use tobacco products at higher rates than women do, and some authors suggest that smoking behavior differs substantially between men and women due to education, marital status, and employment conditions [88]. Therefore, the correlation between miRNA expression and their target mRNAs with exposure to environmental factors could potentially modulate their roles in NSCLC and COPD according to gender.

## 5. Conclusions

Herein, we present a pilot study to analyze tissue miRNA profiles related to shared pathways of NSCLC and COPD. A limitation of this study was the small sample size for the miRNA analyses. For this reason, we could not stratify the subjects according to their smoking status or gender. Nevertheless, due to the difficulty in obtaining lung samples from COPD subjects, our findings could be clinically relevant and pave the way for future studies involving a larger population. Taken together, these data, even if preliminary, highlight that lung miRNA expression levels might represent a specific signature of a pro-tumorigenic microenvironment, suggesting their possible role as biomarkers for improving the diagnosis and evaluation of susceptibility to NSCLC in COPD patients. In particular, monitoring miRNAs whose expression is opposite between diseases could provide a proportion of risk to cancer development in subjects with COPD. For instance, an increase in the levels of 149-3p and 197-3p or a decrease in hsa-miR-34a-5p levels could represent the first sign of the establishment of an environment that is more prone to tumorigenesis. Moreover, a direct correlation between miRNAs and their cancer-related targets could clarify the molecular mechanisms underlying the process of carcinogenesis. Therefore, further investigations in a larger cohort are needed to understand the exact role of miRNA signatures in COPD, how disease severity can lead to a significant alteration in miRNAs, and mRNA’s impact on NSCLC onset. 

## Figures and Tables

**Figure 1 biomedicines-11-00736-f001:**
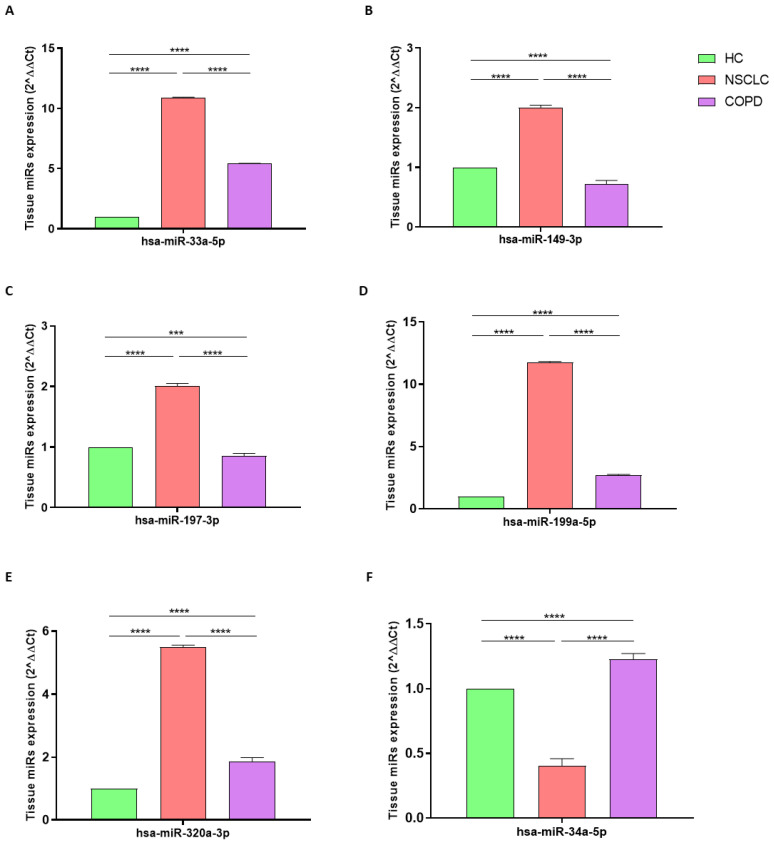
Analysis of hsa-miR-33a-5p (**A**), hsa-miR-149-3p (**B**), hsa-miR-197-3p (**C**), hsa-miR-199a-5p (**D**), hsa-miR-320a-3p (**E**), and hsa-miR-33a-5p (**F**) tissue expression levels in HC (green column; biological replicates n = 6), NSCLC (pink column; biological replicates n = 9). and COPD (purple column; biological replicates n = 6). All samples were run in triplicate, and results are shown as means ± SD. The statistical tests used in these analyses were one-way analysis of variance followed by Dunn’s Multiple Comparison Test. *** *p* < 0.001; **** *p* < 0.0001.

**Table 1 biomedicines-11-00736-t001:** Demographic characteristics of the enrolled patients. Data are means ± SD unless specified. The statistical tests used in these analyses were one-way ANOVA followed by the Tukey Multiple Comparison Test. FEV1: forced expiratory volume in the first second; FVC, forced vital capacity; LABA: long-acting beta-agonists; SABA: short-acting beta-agonists; LAMA: long-acting muscarinic agents.

	Healthy Control (HC)	NSCLC	COPD	*p*-Value
**Total Participants (N)**	6	9	6	
**Age (SD)**	70.3 (7.27)	76.7 (3.82)	65.0 (11.31)	0.0001
**Gender (M/F)**	5/1	7/2	3/3	0.0097
**Smoking history (pack years)**	5.0	21.5	17	<0.0001
**Current smoker**	1	3	4	0.0481
**Former smoker**	2	5	0	<0.0001
**FEV1 % predicted (SD)**	91% (16)	94% (19)	67% (16)	<00001
**FEV/FVC (SD)**	88 (5)	78 (4)	62 (3)	<0.0001
**Comorbidities**				
**Arterial Hypertension (%)**	3 (50.0%)	4 (44.4%)	2 (33.3%)	0.0001
**Other cardiovascular diseases (%)**	1 (16.6%)	4 (44.4%)	1 (16.6%)	<0.0001
**Diabetes Mellitus (%)**	2 (33.3%)	5 (55.5%)	1 (16.6%)	<0.0001
**Medications**				
**Inhaled corticosteroids (N, %)**	0	1 (11.11%)	1 (16.6%)	<0.001
**LABA/SABA/LAMA (N, %)**	0	0 (0	3 (50%)	<0.0001

**Table 2 biomedicines-11-00736-t002:** Bioinformatics tools for in silico analysis. Number of validated genes for each miRNA analyzed in miR Target Link Human and Diana Tools databases.

Number of Target Genes		
miRNA	miR Target Link Human	DIANA Tools
hsa-miR-33a-5p	191	713
hsa-miR-149-3p	1274	1469
hsa-miR-197-3p	559	1004
hsa-miR-199a-5p	280	666
hsa-miR-320a-3p	841	1946
hsa-miR-34a-5p	615	1108

**Table 3 biomedicines-11-00736-t003:** Abbreviations, gene names, methods of validation, and tissues on which the miRNAs selected were validated targets from DIANA Tools database. Bi: Biotin; IP: Immunoprecipitation; MA: Microarrays; qP: qPCR; RA: Reporter Gene Assay; O: Other; WB: Western Blot.

Abbreviation	Gene Name	Methods	Tissue
*ADAM10*	A disintegrin and metalloproteinase 10	Bi	Intestine
*XBP1*	X-Box Binding Protein 1	Bi, IP, MA	Intestine, Bone Marrow, Kidney
*ZNF281*	Zinc Finger Protein 281	IP	Kidney, Bone Marrow
*CDK6*	Cyclin-Dependent Kinase 6	Bi, MA, IP, qP, RA, O, WB	Intestine, Bone Marrow, Kidney, Liver
*LATS1*	Large Tumor Suppressor Kinase 1	IP	Pancreas, Cervix
*LATS2*	Large Tumor Suppressor Kinase 2	IP	Cervix
*MARCH8*	Membrane Associated Ring-CH-Type Finger 8	IP	Pancreas
*MARCH9*	Membrane Associated Ring-CH-Type Finger 9	IP	Mammary Gland
*SMAD7*	SMAD Family Member 7	IP	Cervix
*E2F3*	E2F Transcription Factor 3	Bi, MA, IP, QP, WB	Kidney, Embryo, Cervix, Intestine, Bone Marrow
*CAPRIN 1*	Cytoplasmic activation/proliferation-associated protein-1	IP	Pancreas
HEATR1	HEAT repeat-containing protein 1	IP	Bone Marrow

**Table 4 biomedicines-11-00736-t004:** miRNA gene interaction and possible biochemical pathways involved.

	Biochemical Pathways	
	miRNA	Validated Target Genes
NOTCH1 pathway; cell migration and invasion	hsa-miR-34a-5p	ADAM10
		XBP1
NF-kappa B pathway; inhibition of cell apoptosis and promotion of cell proliferation; poor overall survival	hsa-miR-33a-5p hsa-miR-320a-3p	ZNF281
	hsa-miR-149-3p	HEATR1
Cell cycle deregulation and uncontrolled cell proliferation	hsa-miR-34a-5p hsa-miR-199a-5p	CDK6 CAPRIN1
JNK, GPCR, and Hippo pathways; loss of tumor suppressor capability	hsa-miR-320a-3p	LATS1 LATS2
PI3K, mTOR, and ICAM-1 signaling pathways; poor clinical outcomes	hsa-miR-199a-5p	MARCH8
	hsa-miR-197-3p	MARCH9
TGFβ signaling pathway; promotion of epithelial–mesenchymal transitions and metastasis	hsa-miR-33a-5p	SMAD7
Bcl-2 signaling pathway; uncontrolled tumor growth	hsa-miR-34a-5p	E2F3

## Data Availability

Data sharing not applicable.

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
