# Peer review of "Lung microRNAs Expression in Lung Cancer and COPD: A Preliminary Study"

_biomedicines, 2023, doi:10.3390/biomedicines11030736_

Round 1

Reviewer 1 Report

This study compared miRNA expression in NSCLC, COPD, and healthy control participants. The authors attempted to uncover microRNA (miRNA) signatures in lung tissue that may be related with non-small lung cancer (NSCLC). The authors gathered lung samples and assessed the expression of miRNA using RT-PCR. They discovered a distinct pattern in the expression of the miRNAs hsa-miR-149-3p, 197-3p, and 34a-5p in NSCLC, pointing to their potential function as markers of the tumor microenvironment. The in-silico research also discovered brand-new targets, including those connected to carcinogenesis, like members of the Cyclin Dependent Kinase family. Overall, the work offers proof that miRNA signatures may be used to enhance the assessment of lung cancer initiation. The results are reported clearly and succinctly in the well-organized manuscript. The authors have presented a comprehensive analysis and have successfully used data to support their conclusions. The findings are reliable and important. Researchers studying miRNA and cancer biology will find this material interesting and I recommend it for publication.

Minor points:

1.       It seems that the procedure outlined for the RT-PCR study of miRNA expression is thorough. However, more details regarding the quality control procedures used to guarantee the integrity and purity of the RNA samples prior to reverse transcription and the normalization technique applied to account for variability in the RT-PCR reactions would be helpful.

2.       Figure 5, 6: The Y-axis needs to be rescaled.

Author Response

This study compared miRNA expression in NSCLC, COPD, and healthy control participants. The authors attempted to uncover microRNA (miRNA) signatures in lung tissue that may be related with non-small lung cancer (NSCLC). The authors gathered lung samples and assessed the expression of miRNA using RT-PCR. They discovered a distinct pattern in the expression of the miRNAs hsa-miR-149-3p, 197-3p, and 34a-5p in NSCLC, pointing to their potential function as markers of the tumor microenvironment. The in-silico research also discovered brand-new targets, including those connected to carcinogenesis, like members of the Cyclin Dependent Kinase family. Overall, the work offers proof that miRNA signatures may be used to enhance the assessment of lung cancer initiation. The results are reported clearly and succinctly in the well-organized manuscript. The authors have presented a comprehensive analysis and have successfully used data to support their conclusions. The findings are reliable and important. Researchers studying miRNA and cancer biology will find this material interesting and I recommend it for publication.

  1. We would like to thank reviewer for comments and remarks. We hope that the revision of our manuscript according to the reviewers´ comments and our answers (in blue) are further improving our manuscript.

Minor points:

  1. It seems that the procedure outlined for the RT-PCR study of miRNA expression is thorough. However, more details regarding the quality control procedures used to guarantee the integrity and purity of the RNA samples prior to reverse transcription and the normalization technique applied to account for variability in the RT-PCR reactions would be helpful.
  2. As request, we have added more information about quality control procedures. RNA samples integrity and purity were assessed by the qubit RNA Integrity and Quality (IQ) assay using a Qubit 4 fluorometer. Additionally, we selected U6 snRNA (Qiagen) as housekeeping gene to perform normalization. We have acknowledged this information in the “Materials and Methods” section.
  3. Figure 5, 6: The Y-axis needs to be rescaled.
  4. We have modified the Y-axis, as requested.

Reviewer 2 Report

In the manuscript “Lung microRNAs expression in lung cancer and COPD: a preliminary study”, Mirra et al investigated the potential differential expression of the miRNAs among NSCLC, COPD, and healthy controls in human lung specimens. They identified several differential expressed miRNAs and explore their potential targets based on in-silico analysis. The current work is quite simple, as the title shows “preliminary study”.  The following major revisions need to be considered before its consideration for publication.

Major comments

1. How many samples were used in the Figures 1-6? The variation between experiments is very low and must be clarified that biological replicates are made. In general, the technical replicate is discouraging to perform the statistical analysis. A clear definition should be added to the legends of relevant figures.

2. The Figure 1-6 should be merged into one figure.

3. In the Table 1, the decimal point should be represented by full stop (.), but not comma (,).

4. Why the authors directly focused on hsa-miR- 33a-5p, 149-5p, 199a-5p, 320a-3p and 197-3p, but not other miRNAs? Is there any screening process? The logical behind this should be well explained with evidence from references.

5. Total RNA was extracted from paraffin embedded samples; the extraction efficiency and RNA integrity should be affected. Did the authors validate the expression of the miRNAs they chosen based on the public datasets, such as TCGA and published literatures?

6. The primers used in this study should be provide.

Minor comments

miRs should be miRNAs

Author Response

In the manuscript “Lung microRNAs expression in lung cancer and COPD: a preliminary study”, Mirra et al investigated the potential differential expression of the miRNAs among NSCLC, COPD, and healthy controls in human lung specimens. They identified several differential expressed miRNAs and explore their potential targets based on in-silico analysis. The current work is quite simple, as the title shows “preliminary study”.  The following major revisions need to be considered before its consideration for publication.

  1. We would like to thank reviewer for comments and remarks. We hope that the revision of our manuscript according to the reviewers´ comments and our answers (in blue) are improving our manuscript.

Major comments

  1. How many samples were used in the Figures 1-6? The variation between experiments is very low and must be clarified that biological replicates are made. In general, the technical replicate is discouraging to perform the statistical analysis. A clear definition should be added to the legends of relevant figures.
  2. For each miRNA 18 samples were used: “HC”; n = 6; “NSCLC”; n=9; “COPD,” n =6 and all samples were run in triplicate. We have acknowledged this information in figures legends.

  1. The Figure 1-6 should be merged into one figure.
  2. As requested, we have merged the figure 1-6 into one figure.

  1. In the Table 1, the decimal point should be represented by full stop (.), but not comma (,).
  2. We have modified Table 1.

  1. Why the authors directly focused on hsa-miR- 33a-5p, 149-5p, 199a-5p, 320a-3p and 197-3p, but not other miRNAs? Is there any screening process? The logical behind this should be well explained with evidence from references.
  2. We thank the reviewer for his remark. As previously reported, all the selected miRNAs were chosen from the literature data. Specifically, we focused on those miRNAs that had been shown to influence cancer hallmarks, such as increased cell proliferation or the establishment of an immunosuppressive environment. We have extended the 'Introduction’ section and references to better clarify our choice.
  3. Total RNA was extracted from paraffin embedded samples; the extraction efficiency and RNA integrity should be affected. Did the authors validate the expression of the miRNAs they chosen based on the public datasets, such as TCGA and published literatures?
  4. We recognize that RNA integrity from paraffin embedded samples can be affected. Indeed, RNA integrity and purity were assessed by the qubit RNA Integrity and Quality (IQ) assay using a Qubit 4 fluorometer. We apologize to the reviewer for not having added this information to the manuscript previously therefore, we have now added it in the in the “Materials and Methods” section. All selected miRNAs were chosen from the public literature as reported in the 'Introduction' and 'Discussion' sections. In addition, their target mRNAs were selected from public databases such as “DIANA tools” and “miR target link human” and possible biochemical pathways involved were checked using GeneCard database, as we reported. All mRNAs’ targets chosen were selected from those validated by techniques such as immunoprecipitation or western blot.

  1. The primers used in this study should be provide.

The primer used were custom Stem-loop primer, Thermo Fisher Scientific (Waltham, MA, USA).We have added primers sequences in the “Materials and Methods” section.

Minor comments

miRs should be miRNAs

  1. We have corrected it.

Reviewer 3 Report

Mirra et al, studied expression of few miRNAs in lung tumors and COPD samples and compared with healthy controls. The study lack rationale of designing experiment and needs clarity.

Comments:

1. Rationale of studying expression of selected miRNAs?

2. Significant breakthrough research has been done in cancer for early detection, precision medicines in lung cancer specially for EGFR, ALK , KRAS mutants etc. I am not sure why authors have used vague statements in manuscript like " Despite increasing attention, the mechanisms underlying tumorigenesis remain unclear, making novel biomarkers necessary to improve early diagnosis and prediction of progression".

3. Mutation profile available for lung cancer patients studied. Do they have any oncogenic mutation or history of pretreatment.

4. If patients have some oncogenic mutation at first place what is the rationale of studying miRNAs?

5. 149-5p or 3p? text does not match with figures

6. It would be interesting to biology of miRNAa like 332-5p and 199a-5p, 320a-3p? They are gradually increasing from HC to COPD to cancer. May be they dictate some common biological phenotype in COPD and cancer.

6. In-silico and pathway analysis need more explanation. 

Author Response

Mirra et al, studied expression of few miRNAs in lung tumors and COPD samples and compared with healthy controls. The study lack rationale of designing experiment and needs clarity.

  1. We would like to thank reviewer for comments and remarks. We hope that the revision of our manuscript according to the reviewers´ comments and our answers (in blue) are improving our manuscript.

Comments:

  1. Rationale of studying expression of selected miRNAs?
  2. We thank the reviewer for his remark. As previously reported, all the selected miRNAs were chosen from the literature data. Specifically, we focused on those miRNAs that had been shown to influence cancer hallmarks, such as increased cell proliferation or the establishment of an immunosuppressive environment. We have extended the 'Introduction’ section and references to better clarify our choice.

  1. Significant breakthrough research has been done in cancer for early detection, precision medicines in lung cancer specially for EGFR, ALK , KRAS mutants etc. I am not sure why authors have used vague statements in manuscript like " Despite increasing attention, the mechanisms underlying tumorigenesis remain unclear, making novel biomarkers necessary to improve early diagnosis and prediction of progression".
  2. We agree with the reviewer regarding the relevance of newly identified biomarkers for lung cancer early detection like EGFR, ALK or KRAS mutation. Our statement particularly referred to tumorigenesis in COPD patients, which still remain unclear and needs further in-depth analysis. Therefore, we have introduced a more appropriate statements within the manuscript.

  1. Mutation profile available for lung cancer patients studied. Do they have any oncogenic mutation or history of pretreatment.
  2. If patients have some oncogenic mutation at first place what is the rationale of studying miRNAs?
  3. We want to thank the reviewer for this comment. We do see the valid point of performing experiments in this direction. However, at this stage, studying relatively unknown mutation profiles of which we do not know if they can correlate with the miRNAs chosen does not fit our aims. In any case, we would like to study this deeply since we recognize the suggestion as an exciting starting point, but this would be the subject of a separate future manuscript. Finally, none of the patients enrolled had received any pharmacological treatment, as biopsy tissue was collected during bronchoscopy to confirm the diagnosis.

  1. 149-5p or 3p? text does not match with figures
  2. We apologize for confusing the reviewer by typing the wrong miR ID. We have entered the correct caption.

  1. It would be interesting to biology of miRNAa like 332-5p and 199a-5p, 320a-3p? They are gradually increasing from HC to COPD to cancer. May be they dictate some common biological phenotype in COPD and cancer.
  2. We agree with the reviewer on the importance of miRNA like 33, 199 and 320a whose similar expression between lung cancer and COPD highlights a possible common microenvironment. Moreover, we think that miRNA like 34, 149 and 197 could be more interesting due to their opposite trend in lung cancer and COPD. Indeed, as described in ‘discussion’ section, monitoring of these latter miRNAs in COPD subjects could be helpful as biomarkers for improving the diagnosis and evaluation of susceptibility to lung cancer in COPD patients.
  3. In-silico and pathway analysis need more explanation.
  4. As request, we have added in the in the “Materials and Methods” section more details about in-silico analysis and database used to obtain information on possible pathways involved.

Round 2

Reviewer 2 Report

Thanks for the author’s reply. The authors have already solved parts of my previously comments, and the revised version has been improved to a certain extent. But the draft needs a more professional and careful modification before considering publication. There are still several comments.

The figures should be re-organized with professional software, for example, Adobe Illustrator. Line weight, color, font type, font size, and font color should be unified and keep them looking good.

The variation between experiments is very low and must be clarified that biological replicates are made.

Author Response

Thanks for the author’s reply. The authors have already solved parts of my previously comments, and the revised version has been improved to a certain extent. But the draft needs a more professional and careful modification before considering publication. There are still several comments.

  1. We thank the reviewer for his observation, and following his suggestions, we have further modified the manuscript to make the data presentation clearer.

The figures should be re-organized with professional software, for example, Adobe Illustrator. Line weight, color, font type, font size, and font color should be unified and keep them looking good.

  1. As request, we have edited all figures and improved their appearance.

The variation between experiments is very low and must be clarified that biological replicates are made.

  1. Following the reviewer's suggestion, we have better explained this information in the “Materials and Methods section - 2.3. Real Time PCR (RT-PCR) paragraph” and in the figure 1 legend.

Reviewer 3 Report

The manuscript can be accepted in present form. 

Author Response

The manuscript can be accepted in present form.

  1. We thank the reviewer for his remarks and suggestions and for helping us improve our manuscript.

Round 3

Reviewer 2 Report

Thanks for the author’s reply. The authors have already solved all of my comments.